# Feasibility and preliminary effects of a mindfulness-based physical exercise (MBPE) program for community-dwelling older people with sarcopenia: A protocol for a parallel, two-armed pilot randomised controlled trial

**Meng-Li Li[1], Patrick Pui-Kin Kor[1]\*, Zhi-Ying Zhang[2], Justina Yat-Wa Liu[1]**

1 School of Nursing, The Hong Kong Polytechnic University, Hong Kong, China, 2 Department of Mood Disorders, Soochow University Affiliated Guangji Hospital, Jiang Su, China

\* patrick.kor@polyu.edu.hk

## Abstract

### Introduction

Physical exercise (PE) is essential for alleviating the symptoms of sarcopenia. Low motivation is a major barrier to PE. Mindfulness-based intervention (MBI) has the potential to improve motivation. However, few studies have used a mindfulness-based PE (MBPE) intervention among older people with sarcopenia.

### Objectives

To assess the feasibility, acceptability and preliminary effects of the MBPE program among community-dwelling older people with sarcopenia.

### Methods and analysis

A two-arm pilot randomised controlled trial will be conducted to assess the feasibility, acceptability and preliminary effects of an MBPE program among community-dwelling older people with sarcopenia. A total of 60 participants will be randomised into the intervention group, receiving the MBPE intervention twice a week over 12 weeks, or the control group, receiving health education with the same duration, number of sessions and frequency as the intervention group. Each session of the MBPE program will last about 60 min, including 5-10- min introduction, 20-min MBI, 30-min PE and 5-10-min sharing and discussion. The primary outcomes will be the feasibility (i.e., the time spent recruiting participants, the eligibility rate and the recruitment rate) and acceptability (i.e., the attendance rate, completion rate and attrition rate) of the MBPE program. The secondary outcomes will be the preliminary effects of the MBPE program on symptoms of sarcopenia, motivation for PE, psychological well-being, mindfulness level, physical activity level and quality of life. Individual

**Data Availability Statement:** No datasets were generated or analysed during the current study. All relevant data from this study will be made available upon study completion.

**Funding:** The author(s) received no specific funding for this work.

**Competing interests:** The authors have declared that no competing interests exist.

interviews will be conducted to identify the strengths, limitations and therapeutic components of the intervention. The quantitative data will be analysed by generalised estimating equations. The qualitative data will be analysed by Braun and Clarke's thematic approach.

## Conclusion

The findings of this study will be able to provide evidence for the health professionals in adopting MBPE as a supportive intervention for the older adults with sarcopenia and the groundworks for the researchers in developing non-pharmacological intervention for older adults. The positive effects could facilitate healthy ageing and relief the burden of the medical system, especially in the countries facing the ageing population.

## Trial registration number

NCT05982067; ClinicalTrials.gov.

## Introduction

Sarcopenia is a progressive and generalised skeletal muscle disorder characterised by decreased muscle mass, muscle strength and physical function [1]. It is classified as an independent disease with the code ICD-10-CM (M62.84) by the World Health Organization [2]. Globally, it is estimated that over 120 million older adults are living with sarcopenia, and this number is expected to double by 2050 [3, 4]. Compared with healthy ageing, sarcopenia significantly increases the risks of various adverse health events, such as falling by 3.23 times, fracture by up to 3.75 times [5], hospitalisation by 2.07 times [6] and all-cause mortality by 2.20 times [7] and it is also negatively associated with psychological well-being [8]. Sarcopenia greatly increases healthcare costs and burden in medical system [9]. Thus, it is imperative to find a way to manage sarcopenia in order to promote healthy ageing worldwide.

Currently, there are no approved medications for treating sarcopenia [10]. Lifestyle interventions, especially physical exercise (PE) interventions, have been demonstrated to mitigate the symptoms of sarcopenia [11–15]. Thus, PE is strongly recommended as the first-line treatment [16]. However, the low adherence and motivation to regular PE is a major concern in older adults with sarcopenia [17]. A systematic review (SR) demonstrated that the adherence to PE interventions by healthy older adults was approximately 60% [18]. The adherence of older adults with sarcopenia might be even lower than 60% due to their decreased physical function, PE ability and mobility [19]. This problem has been extensively investigated in numerous prior studies [20–23], but effective interventions and management have not been adequately addressed or explored in the existing literature.

Distraction, mind-wandering and negative thoughts and feelings, such as fatigue, anxious or hopeless during the PE, were the common barriers affecting the motivation among the older adults with sarcopenia in performing PE, in addition to physical function and supervision [24, 25]. For example, study has indicated that older adults with sarcopenia are easily derailed by issues in daily life [26] and misled by their preconceived misunderstandings about PE, such as the belief that they are too old to perform any kind of PE [20]. The PRIME (Plans, Impulses, Inhibitions, Motivations, Evaluations and Response) theory of motivation (S1 Fig, details in Conceptual framework of the MBPE program section) explained diversity of factors that influence motivation and emphasized the positive feelings and non-judgmental self-awareness for engaging people in goal-directed behaviour [25, 27]. Although prior studies

adopted various strategies to improve adherence and motivation to PE for the older adults with sarcopenia [12], only a few of them took these components (e.g., distraction, negative feelings) into consideration when designing the PE program. Moreover, as the above negative feelings and experiences of older adults with sarcopenia regarding PE are also likely to undermine their psychological well-being and may lead to mental health issues (e.g., anxiety, depression) [28]. Therefore, a way to promote the positive feelings and non-judgmental self-awareness to enhance their motivation in performing PE and ultimately improve their physical health as well as psychological well-being is urgently required.

Mindfulness is defined as a process of openly attending with awareness to one's present moment [29]. Mindfulness-based intervention (BMI) is primarily designed for treating depression and other mental health problems [30, 31]. Additionally, an increasing number of studies have demonstrated its positive effects in enhancing the PE levels in different populations [32–34]. One SR [35] concluded that MBIs effectively activated the motivation to perform PE and the adherence to PE in older adults, and a 4-week RCT [36] reached a similar conclusion. Moreover, the adherence to MBI can be maintained even after the program has ended [37]. The mindful coping model (S2 Fig) offers a possible explanation of the positive effects in which the MBI direct the participants' attention away from their negative thinking, enhance their engagement in the daily activities and PE and evoking positive feelings [38]. This implies that MBIs could be incorporated into PE interventions to improve both the adherence to PE and the psychological well-being of older adults with sarcopenia.

However, despite the benefits of MBIs, few studies of mindfulness embedded into PE interventions for older adults with sarcopenia older people have been conducted. To the best of our knowledge, only two mindfulness-based PE interventions (MBPE) have been conducted in this population, adopting mindful yoga education [39] and Tai Chi [40]. However, limitations exist in these two studies, focusing solely on women [39] or solely on men aged 85 years or older [40], not assessing muscle mass [39, 40], even though muscle loss is a key symptom of sarcopenia. Importantly, neither of these studies included resistance exercise, which is the most highly recommended type of PE to treat sarcopenia [16]. It is imperative to develop a novel intervention combining MBI and PE (specifically resistance exercise) to activate and improve the motivation for and adherence to PE, mitigate sarcopenia and improve psychological well-being in older adults with sarcopenia. Given the limited evidence in this field, this pilot RCT aims to explore the feasibility, acceptability and preliminary effects of a MBPE intervention in community-dwelling older people with sarcopenia.

### Objectives and hypothesis

The objectives of this pilot RCT are as follows: (1) to assess the feasibility and acceptability of the MBPE program among community-dwelling older people with sarcopenia; and (2) to investigate the preliminary effects of the MBPE program on the symptoms of sarcopenia, motivation for PE, psychological well-being, mindfulness level, physical activity level and quality of life among this population.

We hypothesise that the MBPE program is acceptable and feasible with the potential to improve the motivation, adherence to PE, physical and psychological health of the older adults with sarcopenia.

### Methods and analysis

### Study design

This study will adopt a two-arm pilot RCT design with nested individual interviews. The Standard Protocol Items: Recommendations for Interventional Trials (SPIRIT) guideline [41] is

| | Study Period | | | |
|---|---|---|---|---|
| | Enrolment | Allocation | Post-allocation | Close-out |
| TIMEPOINT | -t1 | 0 | 0 | 12 weeks |
| **ENROLMENT:** | | | | |
| Eligibility screen | x | | | |
| Informed consent | x | | | |
| Allocation | | x | | |
| **INTERVENTIONS:** | | | | |
| Mindfulness-based physical exercise | | | ◆————————◆ | |
| Health education | | | ◆————————◆ | |
| **ASSESSMENTS:** | | | | |
| Socio-demographic data | | | x | |
| Feasibility | | | x | |
| Acceptability | | | x | x |
| Muscle mass | | | x | x |
| Muscle strength | | | x | x |
| Physical function | | | x | x |
| Depressive symptoms | | | x | x |
| Psychological well-being | | | x | x |
| Mindfulness level | | | x | x |
| Quality of life | | | x | x |

**Fig 1. The SPIRIT flow of the mindfulness-based physical exercise program.**

followed in this proposal (Fig 1 and S1–S3 Appendices). The overall flow of the pilot RCT in the form of a Consolidated Standards of Reporting Trials (CONSORT) diagram is shown in Fig 2 [42]. The conceptual framework of this study is shown in Fig 3.

## Study setting

Participants will be recruited from three community health care centres (CHCCs) in Suzhou, Jiangsu province, China. The MBPE intervention will be conducted at one of the CHCCs and at participants' homes.

## Eligibility criteria

This study will focus on community-dwelling older people with sarcopenia. The detailed inclusion and exclusion criteria are given in Table 1.

## Sample size

25 per group will provide sufficient information for planning the proposed subsequent main randomised trial with an anticipated standardised effect size of about 0.2 on the primary

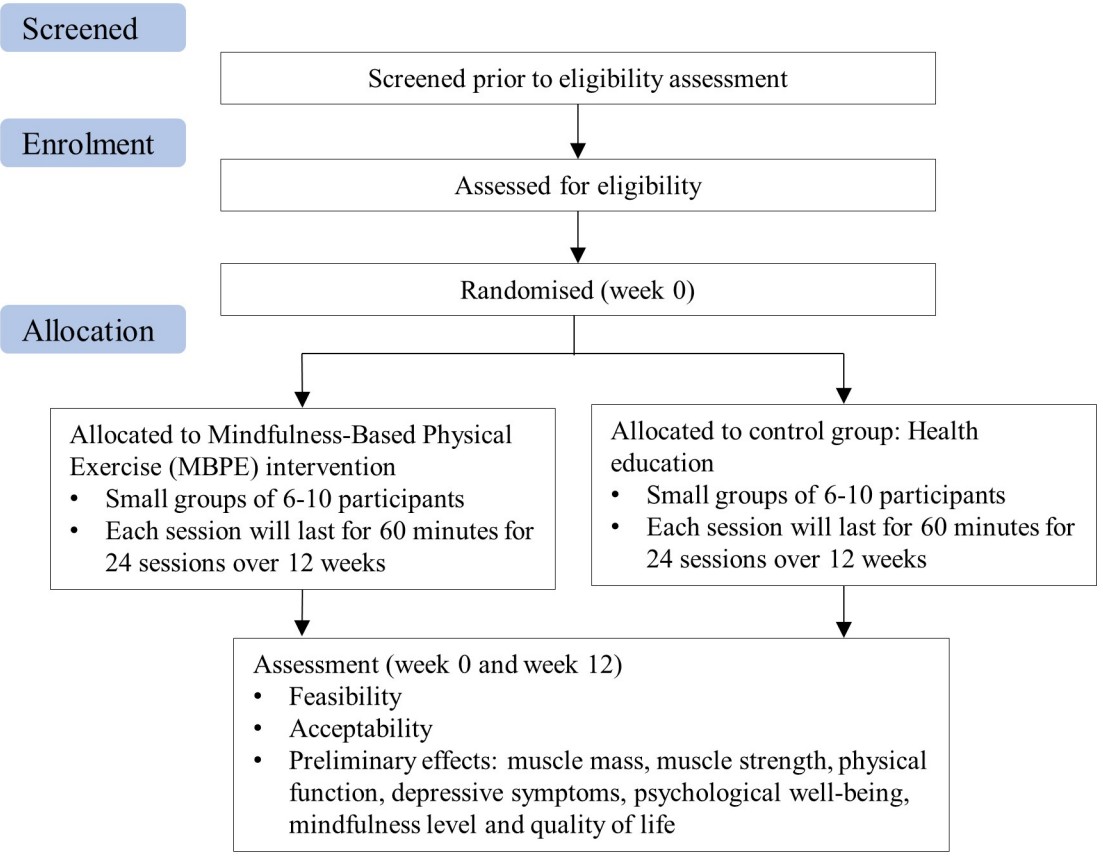

**Fig 2. CONSOROT diagram of the mindfulness-based physical exercise program.**

outcome, handgrip strength [46]. Considering a 20% attrition rate [47], the sample size of this pilot RCT has been calculated as $25 \times (1 + 20\%) = 30$ in each group.

## Allocation and blinding

After they give their informed consent, eligible participants will be randomised into the intervention group (IG) or the control group (CG) with a 1:1 ratio. Block randomisation with a block size of 4 will be used to balance the numbers in each group. Three research assistants who will not otherwise be involved in this study will perform the randomisation independently to ensure allocation concealment and avoid selection bias. The random numbers will be generated by a computer and sealed in opaque envelopes and delivered to each participant.

Due to the characteristics of the interventions (the MBPE vs health education), it will be impossible to blind the participants and interventionists. The following personnel will, however, be blinded: the research assistants recruiting participants, the research assistants performing randomisation and the outcome assessor.

## Intervention development

The detailed development of the MBPE program follows the Medical Research Council (MRC) guideline for complex interventions development [48], which considered the findings from our SR [49], guidelines on PE for older adults with sarcopenia [19, 50], MBIs provided by

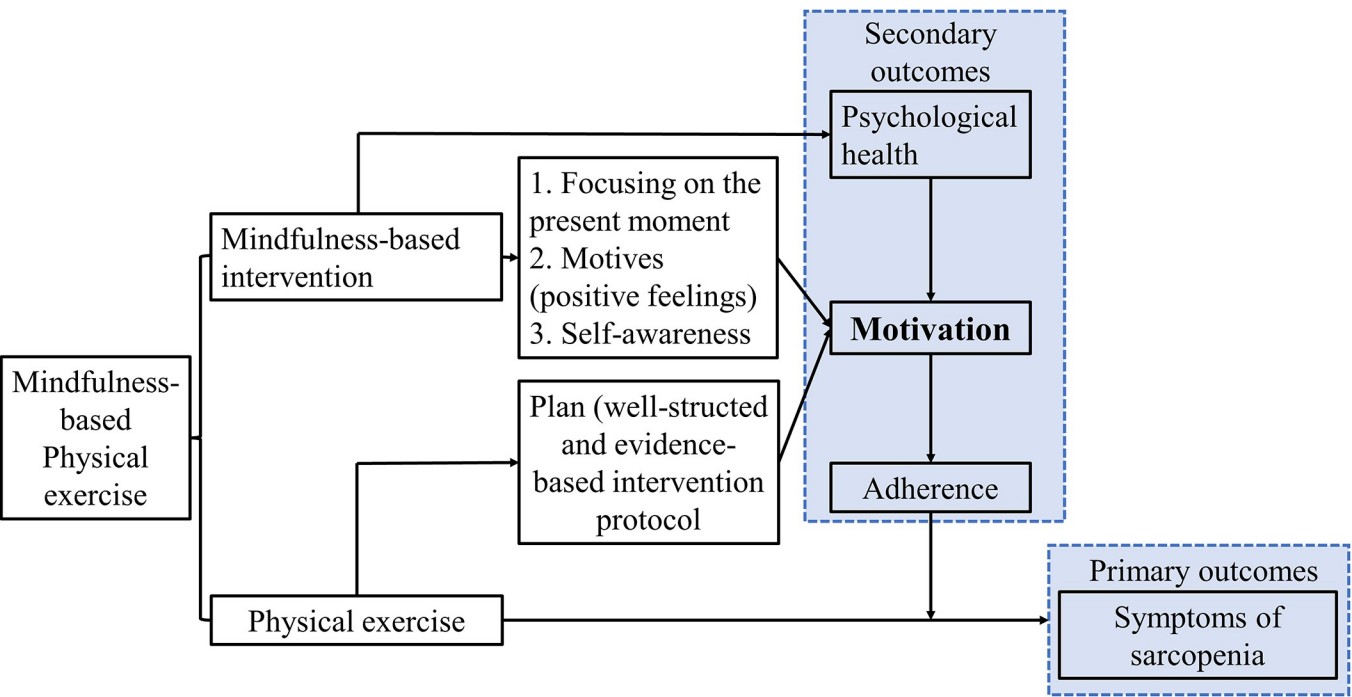

**Fig 3. The conceptual framework of the mindfulness-based physical exercise program.**

the Oxford Mindfulness Centre [51] and Bangor University [52]. After the initail version of the MBPE program, a panel of experts including nurses, mindfulness therapist, researchers in gerontology, and geriatrician validated the protocol via a two-round Delphi process (unpublished yet). The comments and suggestions from the older adults with sarcopenia were collected to refine the MBPE protocol in this proposed study. The study will be conducted in a hybrid mode, combination of at CHCC and at home [53, 54].

**Table 1. Eligibility criteria for participants.**

| **Inclusion criteria** |
| --- |
| 1. Community-dwelling and aged 60 years or older. |
| 2. Diagnosed with sarcopenia according to the criteria of the Asian Working Group for Sarcopenia (AWGS) [43]: <br> a)  decreased muscle strength: handgrip strength of men < 28 kg; handgrip strength of women < 18 kg; <br> b)  or decreased physical performance: 5-time chair-stand test time $\geq$12 s; <br> c)  or decreased muscle mass: skeletal muscle mass index (SMI) of men < 7.0 kg/m$^2$; SMI of women < 5.7 kg/m$^2$. |
| 3. Able to communicate orally and in writing and understand instructions. |
| **Exclusion criteria** |
| 1. Hospitalisation lasting for more than 5 days in the 3 months prior to recruitment. |
| 2. Unable to undergo body composition test, due to having a heart pacemaker, vascular stent, steel plates or other materials in the body. |
| 3. Contraindications to PE, such as severe musculoskeletal disorders, severe cardiovascular diseases or spinal nerve injury. |
| 4. Already undertaking regular PE: 150 min of moderate-intensity activity or 75 min of vigorous-intensity activity per week in the 3 months prior to recruitment [44]. |
| 5. Practising mindfulness/yoga for >45 min a week in the 6 months prior to recruitment [45]. |

## Conceptual framework of the MBPE program

Two theories, the PRIME theory of motivation (S1 Fig) and the mindful coping model (S2 Fig), was adopted to establish the conceptual framework of the MBPE program (Fig 3). Based on the mindful coping model, the mindfulness component in the MBPE program can enhance the motivation for PE by enhancing three essential elements of human motivation in the PRIME theory of motivation: focusing on the present moment, accepting of negative feelings (Self-awareness and Motives), and evoking positive feelings (Motives), which eventually maximize the benefits of PE and improve the physical health of the participants. In addition, the mindful coping model indicates that the mindfulness practice can improve the psychological health of older adults with sarcopenia.

## Intervention description

**1. The MBPE intervention in the IG.**    The MBPE intervention aims to help participants learn mindfulness and maintain the state of mindful during PE to improve their experiences of PE and motivation to perform PE [27, 38]. Each session will last for approximately 60 min and have a concrete theme, learning objectives and corresponding activities, including a 5–10-min introduction, 20-min MBI, 30-min PE and 5–10-min sharing and discussion [49, 51, 52]; details are provided in Table 2. The contents and activities of the MBI components will be adapted to the theme of each session, whereas the PE movements will be the same throughout the program but with increased intensity every two weeks until a moderate intensity (measured using the Borg scale) is achieved [50]. The MBPE intervention will be conducted twice a week for 12 weeks [19]. The content of week 8 will be repeated in weeks 9–12 to enhance the benefits of MBPE and help the participants incorporate MBPE into their daily lives, given that they are expected to be able to learn the MBI within 8 weeks [54]. To better combine the MBI and PE components, tips on remaining mindful during PE will be given to the participants.

To facilitate the participants' learning, the face-to-face sessions will be provided in the first four weeks under the supervision of a qualified mindfulness therapist and a qualified sports coach in small groups of 6–10 participants to ensure adequate supervision at the CHCC [55]. Then, the intervention sessions will gradually transition to hybrid mode during weeks 5–12. Specifically, the participants will attend the face-to-face sessions once a week in weeks 5–8 and once every two weeks in weeks 9–12. To achieve a session frequency of twice a week, in addition to the sessions at the CHCC, the participants will perform the interventions at home using videos and templates containing the MBIs and PE components of our MBPE intervention.

**2. Health education in the CG.**    The aim of the CG will be to avoid the influence of socialisation and interaction with others on the participants' motivation to perform PE and thus isolate the effectiveness of the MBPE program. Thus, health education consisting of discussions provided by a registered nurse will be conducted. The number of sessions, duration, frequency, group size and delivery modality of the CG will be similar to those of the IG. The topics of this health education will be the prevention and management of common diseases among older adults, such as hypertension and diabetes, and will not include MBI or PE.

## Primary outcomes and measures

The primary outcomes of this pilot RCT will be the feasibility and acceptability of the MBPE intervention.

**1. Feasibility of the MBPE intervention.**    Feasibility will be evaluated using the time spent recruiting participants, the eligibility rate (number of eligible participants / number of

**Table 2. Themes, mindfulness and PE components in the MBPE intervention.**

| Week | Theme of each session | Objectives | MBI component (20 min) | PE component (40 min) |
|---|---|---|---|---|
| 1 | Introduction to MBPE | Help participants to understand mindfulness, MBPE and key points of keeping mindful | 1) Mini lectures on mindfulness, MBPE and key points of remaining mindful during PE 2) Initial experience of mindfulness (3-min breath meditation) | **PE movement** 1) Warm up (10 min): mindful torso rotation, mindful shoulder rotation, mindful wrist wrap, mindful knee circles, mindful ankle rotation 2) RE (20 min): mindful lateral hand raises, mindful hip abduction, mindful bicep flexors, mindful straight leg lift, mindful handgrip strength exercise, mindful toes up, mindful heels up 3) Cool down (10 min): mindful bicep stretch, mindful tricep stretch, mindful calf stretch, mindful quadricep stretch, mindful cross-legged twist, mindful back stretch **PE intensity** 1) Fairly light (10–11 on Borg scale) 2) 2 sets, 8–10 repeats |
| 2 | Paying attention to breath during static and dynamic movements | Help participants to understand the characteristics of mindfulness and learn how to pay attention to breath | 1) Mini lectures on the key characteristics of mindfulness and the importance of focusing on breath, especially for older people with sarcopenia 2) Mindful practice: 6-min breath meditation 3) Tips on paying attention to breath during PE | **PE movement** Same as Week 1 **PE intensity** Same as Week 1 |
| 3 | Paying attention to body sensations during static and dynamic movements | Help participants to understand the importance of paying attention to the body and know how to do so | 1) Mini lectures on body sensations and appreciation 2) Mindful practice: 10-min body scan 3) Tips on paying attention to body sensations during PE | **PE movement** Same as Week 1 **PE intensity** 1) Somewhat hard (11–12 on Borg scale) 2) 2 sets, 10–12 repeats |
| 4 | Working with pain, challenges, and stress | Help participants to learn to notice pain, challenges and stress in the body and to observe their physical limitations non-attachedly | 1) Mini lecture on pain and stress, especially those that older adults with sarcopenia commonly have in daily life 2) Mindful practice of noticing and observing pain, challenges and stress: 10-min seated meditation 3) Tips on noticing physical pain, limitations, stress or challenges during PE | **PE movement** Same as Week 1 **PE intensity** Same as Week 3 |
| 5 | Working with thoughts during stillness and PE | Help participants to understand the relationship between thoughts and facts and learn to notice thoughts non-judgementally | 1) Mini lecture on the conditioned mind: using some common thoughts of older adults with sarcopenia to elaborate on the topic 2) Awareness of our thoughts: 7-min sitting meditation 3) Tips on noticing and dealing with thoughts during PE | **PE movement** Same as Week 1 **PE intensity** 1) Hard (12–14 on Borg scale) 2) 3 sets, 8–15 repeats |
| 6 | Understanding habitual thinking patterns during life and PE | Help participants to understand that behaviour, thoughts and feelings are interwoven, and learn to notice the tendency to label things during PE | 1) Mini lectures on becoming unstuck and having a flexible mind: using common negative feelings of older adults with sarcopenia to show how these negative feelings stick in the mind, behaviour, etc. 2) Mindful practice: 7-min sitting meditation 3) Tips on observing the tendency to label things during PE | **PE movement** Same as Week 1 **PE intensity** Same as Week 5 |

*(Continued)*

**Table 2.** (Continued)

| Week | Theme of each session | Objectives | MBI component (20 min) | PE component (40 min) |
|---|---|---|---|---|
| 7 | Self-compassion during life and PE | Help participants to understand self-compassion, how to improve self-compassion and show more appreciation towards PE and life | 1) Mini lectures on compassion vs criticism and nourishing vs depleting activities<br>2) Mindful practice: 7-min loving meditation<br>3) Tips on self-compassion during PE | **PE movement**<br>Same as Week 1<br>**PE intensity**<br>Same as Week 5 |
| 8 | Mindfulness-based PE and mindfulness in life | Help participants to learn to integrate mindfulness and MBPE into daily life and to view things from a beginner's perspective | 1) Mini lecture on incorporating mindfulness and mindfulness-based PE into daily life: what informal mindfulness activities we can do in daily life; when, where and how to do formal mindfulness practice, etc.<br>2) Mini lecture on keeping a beginner's mindset: its importance and how to re-view the things that we ignore or take for granted<br>3) Mindful practice: 7-min sitting meditation with life nourishment | **PE movement**<br>Same as Week 1<br>**PE intensity**<br>Same as Week 5 |

MBI, Mindfulness-Based Intervention; MBPE, Mindfulness-Based Physical Exercise; PE, Physical Exercise.

screened participants) and the recruitment rate (number of participants recruited / number of eligible participants).

**2. Acceptability of the MBPE program.** Acceptability will be evaluated using the prospective acceptability, the concurrent acceptability and the retrospective acceptability [56]. Prospective acceptability will be measured as the recruitment rate and the reasons for which older adults did not get involved in the program[56]. Concurrent acceptability will be assessed using the attendance rate (sessions attended by the participants / all sessions), completion rate (sessions in which participants were actually involved and at least 80% completed / all attended sessions), attrition rate (number of participants dropping out / total number of participants) and the reasons for discontinuing [56]. Attrition will be considered if the completion rate is less than 80%. To have a depth exploration of the participants' experience and avoid leaking privacy, individual interviews will be used to measure the retrospective acceptability, the detailed guideline in Table 3. The individual interviews will be conducted by two researchers who both have experience in individual interviewing and are familiar with sarcopenia and mindfulness. Purposive sampling will be used, and the sample size will depend on the data duration. Reflective journals written by the interviewers will be used to avoid bias and ensure the transparency of the research process [57]. With the participants' consent, the individual interviews will be carried out face-to-face and audio recorded.

### Secondary outcome measures

**1. The symptoms of sarcopenia.** Following the AWGS 2019 [43], the symptoms of sarcopenia include low muscle mass, low handgrip strength and low 5-time chair-stand test scores. Handgrip strength is the primary efficacy outcome as it is the first-line indicator to diagnose sarcopenia [43]. Muscle mass will be assessed via calculating SMI, appendicular skeletal muscle mass, fat-free body weight and total skeletal muscle mass by bioelectrical impedance analysis (Inbody 270, Korea) [58]. Handgrip strength will be measured by a Jamar dynamometer (Jamar, 563213, USA) as recommended by the American Society of Hand Therapists [59]. The maximum value of dominant hand grip strength will be utilised as the result. In the 5-time chair-stand test, the participants will be invited to stand and sit from a chair (approximately 43

**Table 3. Guide to individual interviewing of participants in intervention group.**

| Number | Questions |
|---|---|
| Q1 | **Generally, what do you think of this intervention?** |
| | ***Prompts:*** |
| | -Can you share your feelings about this intervention? |
| | -What impact do you think this intervention will have on your life? |
| Q2 | **What do you think of the content of this intervention?** |
| | ***Prompts:*** |
| | -Which is your favourite part of this intervention? Why? |
| | -Which part do you dislike? Why? |
| Q3 | **What do you think of the frequency, duration and delivery mode of this intervention?** |
| | ***Prompts:*** |
| | -What duration of the intervention do you think is suitable for you? Why? |
| | -What frequency of the intervention do you think is suitable for you? Why? |
| | -What do you think of the delivery mode of this intervention? |
| Q4 | **How about your motivation to exercise?** |
| | ***Prompts:*** |
| | Has it increased? How and why? |
| | Are you feeling confident about doing physical exercise? |
| Q5 | **What recommendations do you have for this intervention?** |
| | ***Prompts:*** |
| | -What do you think should be added? Why? |
| | -What do you think should be revised or deleted? Why? |

cm high) five times and the time taken to complete this exercise will be recorded. Two tests will be conducted with a 1-min interval and the mean of the test results will be taken.

**2. Motivation to exercise.** The motivation to exercise will be evaluated using the Chinese version of the Behavioural Regulation in Exercise Questionnaire-2 (C-BREQ-2), which has been tested in Chinese people and shown good consistency (Cronbach's $\alpha$: 0.72–0.83) [60]. The C-BREQ-2 is a self-report measure with 18 items, divided into five domains: amotivation, external regulation, introjected regulation, identified regulation and intrinsic regulation.

**3. Depressive symptoms.** Depressive symptoms will be measured with the Chinese version of the short-form Geriatric Depression Scale (GDS-15). The GDS-15 has shown acceptable reliability and validity (Cronbach's $\alpha$: 0.793, test–re-test reliability: 0.728) in Chinese older adults [61]. Depressive symptoms are classified based on the score as follows: 0–4: normal; 5–8: mild depression; 9–11: moderate depression; 12–15: severe depression [62].

**4. Psychological well-being.** Psychological well-being will be measured with the Chinese version of Raff's Psychological Well-being Scale, as adapted by Tan [63]. The Cronbach's $\alpha$ of this version is 0.89 and the test–re-test reliability is 0.90.

**5. Physical activity level.** Physical activity level will be evaluated using the Chinese Version of the Physical Activity Scale for the Elderly (PASE-C) [64]. The PASE-C has shown good reliability (intraclass correlation coefficient: 0.81) [65] and validity (fair to moderate association with energy expenditure, walking steps and handgrip strength) [66].

**6. Mindfulness level.** The mindfulness level of the participants will be assessed using the short Chinese version of the Five Facet Mindfulness Questionnaire (FFMQ-15-C). The FFMQ-15-C is a self-rated scale developed by Zhu et al. [67] with acceptable reliability (Cronbach's $\alpha$ of all five facets: $\geq$ 0.73). It includes 15 items rated on a 5-point Likert scale, ranging

from 1 = 'never or very rarely true' to 5 = 'very often or always true' to evaluate the five facets of mindfulness: observing, describing, acting with awareness, non-judging and non-reacting.

**7. Quality of life.** Quality of life will be measured with a questionnaire targeted at sarcopenic older people, namely Sarcopenia and Quality of Life (SarQoL$^®$), which was first developed by Beaudart in 2015 and has been translated into more than 30 languages [68]. The Chinese version of SarQoL$^®$ was introduced and validated by Li with excellent consistency (Cronbach's $\alpha$: 0.867) and test–re-test reliability (0.997) [69].

In addition to socio-demographic information, the outcomes will be evaluated at baseline and immediately after the intervention. The assessors will learn and practice the questionnaires or outcome measurements until the re-test reliability is more than 90% before assessment.

## Data management

To protect the security and confidentiality of the data, the personal information (e.g., name, address, contacts) of the participants will be stored on an encrypted hard disk and only the researchers will have access to it. The paper questionnaires will be locked in the researchers' office. All information collected will only be used for this study and will not be disclosed. A codebook containing information on each variable will be created before the data entry step. All the data will be coded and entered by two researchers independently. Only the core members (MLL, PPKK and JYWL) of this research team can access the data.

## Data monitoring

Throughout the study, the corresponding author (PPKK) will monitor recruitment, intervention and retention with the research team (MLL, JYWL and ZYZ). The research team will meet regularly to plan and evaluate the daily activities of this study. The procedure of recruitment, the intervention content, the preparation of assessment, training of research assistants, data management and data analysis will be prepared and monitored by the research team. Besides, the intervention fidelity will be checked by two research assistants.

## Statistical analysis

**1. Quantitative data analysis.** IBM SPSS 23.0 will be used to statistically analyse the data. For continuous variables, the Shapiro–Wilk test will used to evaluate normality. Mean and standard deviation (SD) will used to describe normally distributed data, especially the outcomes to provide information for calculation of standards effect sizes of the subsequent RCT; otherwise, median, interquartile range ($P_{25}$, $P_{75}$) and range (minimum, maximum) will be used.

For continuous variables, an independent-samples $t$ test will be used if the data are normally distributed; otherwise, the Mann–Whitney $U$ test will be used. Chi-square or Fisher's tests will be used for categorical data. Feasibility and acceptability will be presented in terms of rate or absolute number based on the nature of the data. The generalised estimating equation model will be used to analyse the preliminary effects of this study. The standardised mean difference with corresponding 95% confidence intervals will be presented to describe the mean differences of both primary and secondary outcomes between the intervention group and the control group [70]. All data analyses will be conducted with two-tailed tests with a significance level of $p < 0.05$. Both per-protocol (PP) and intention-to-treat analysis will be conducted. The attention rate >80% will be included in the PP analysis.

**2. Qualitative data analysis.** Braun and Clarke's thematic approach will be used to analyse the qualitative data inductively [71]. Two researchers will listen to the recordings repeatedly to prepare a transcript, perform coding and sort the codes into themes. A final code

framework will be formed by refining and defining the themes repeatedly through group discussion among the two researchers and a third researcher. The data will be continuously analysed until no new themes emerge. The Lincoln and Guba criteria [72] will be followed to improve the rigor of the qualitative analysis.

## Ethics and dissemination

Ethical approval for this study has been obtained from the Institutional Review Board of The Hong Kong Polytechnic University (reference number: HSEARS20221117005). This trial has been registered with ClinicalTrials.gov Protocol Registration (ID: NCT05982067). The principles of the Declaration of Helsinki will be followed: autonomy, non-maleficence and beneficence, and confidentiality.

After being fully informed about this study, prospective participants will independently decide whether to participate, and written informed consent will be obtained from those willing to join. They will be free to choose to withdraw from the study at any time. Their right to access health services at the CHCCs will not be influenced. This study is very unlikely to cause harm to the participants, as the MBPE program is evidence- and theory-based and has been evaluated by experts and older people with sarcopenia. All information collected will be used only for this study and will not be disclosed. When disseminating the MBPE program, only research results that do not contain any private information will be presented.

## Adverse events

For the potential risk, the participants may experience mild muscle soreness during the first 2–3 sessions, which is normal for inactive people starting PE. We will try our best to minimise the occurrence of muscle soreness by fully considering the characteristics of our participants, such as gradually increasing the intensity from low to moderate and separating adjacent sessions by two days to let the participants fully recover. However, if muscle soreness or other discomfort does occur, a professional rehabilitation therapist will be referred for further assessment.

## Patient and public involvement

During the development of the MBPE program, older adults with sarcopenia were invited to give comments and suggestions on the initial MBPE intervention protocol. The evaluation of participants with lived experience helped us to review the MBPE program from their perspectives and further modify the protocol to make it more practical and suitable in real life and more tailored to the target population. Additionally, the participants' opinions on the MBPE program will be collected during and after the pilot RCT to further revise the program in readiness for a full RCT.

## Discussion

Sarcopenia is prevalent in older adults and increases the risks of numerous adverse events [5–8]. Although PE is the key treatment for sarcopenia, adherence to PE is low in older people with sarcopenia, undermining the benefits. This study is the first study to adopt an MBPE intervention, combining an MBI and conventional PE, aiming to maximise the effects of PE via enhancing motivation and adherence.

Although PE is the key strategy to treat sarcopenia, older adults with sarcopenia are more likely to have negative thoughts and feelings towards PE, such as feeling fatigue or discomfort while doing exercise [73], feeling sad or hopeless when they think they are unable to do the

intended PE, and feeling being too old or too dangerous to perform any PE [20]. These negative feelings or experiences greatly diminish their motivation to participate in or maintain PE interventions and their psychological well-being. However, few studies have considered these negative feelings of older adults with sarcopenia in their PE interventions. Based on the PRIME theory of motivation and the mindful coping model [24, 35], MBI can be combined with PE to improve the motivation to perform PE through several means: addressing motives, focusing on the present moment, accepting negative feelings and evoking positive feelings. The ability to be more aware of thoughts, emotions and physical sensations during the PE, can be powerfully for helping participants to disengage from their physically inactive lifestyles or negative feelings towards PE and have positive feelings and experiences [74]. More importantly, MBI can impact the response process of the participants' brain and enable them to change their behaviour from the inside, which may be much superior to behaviour change than some techniques from the outside, such as reminder, goals setting, problem solving, which are like to "push" the participants to change [75].

Considering the limited studies using MBPE in older people with sarcopenia [39, 40], our program will provide evidence on the feasibility, acceptability and preliminary effects of MBPE interventions in this field. For health professionals, the study will provide a potentially effective way to improve motivation and adherence among recipients of PE interventions for sarcopenia. For older people with sarcopenia, this study has the potential to improve their holistic well-being, both physically (in terms of symptoms of sarcopenia) and psychologically.

## Strength and limitation

This study will have several strengths. First, the MBPE intervention protocol has been reviewed by an international panel of experts to maximise the rationality, scientific basis and safety of the program. Second, the evaluation of older adults with lived experience of sarcopenia during program development helped us to tailor the MBPE intervention to the participants and real-life practicalities. Third, the individual interviews after the intervention will allow us to further improve the program in readiness for a full RCT via fully considering the participants' perspectives.

Despite its strengths, this pilot study will have several limitations. First, due to practical limitations (e.g., time, money), the long-term effects of the MBPE intervention will not be explored. Second, the nature of a pilot RCT could undermine the power of statistical tests of the effectiveness of the MBPE program. Thus, a full RCT will be required to accurately test the effectiveness of the MBPE program in future. Finally, as this study will target community-dwelling older people with sarcopenia, the effectiveness of the MBPE program for older adults with sarcopenia in other settings (e.g., nursing homes, hospitals) will remain to be explored.

## Supporting information

**S1 Appendix. The SPIRIT Checklist of the study.**
(DOCX)

**S2 Appendix. The informed consent form.**
(DOCX)

**S3 Appendix. The original study protocol approved by ethics committee.**
(PDF)

**S1 Fig. The PRIME theory of motivation.**
(TIF)

**S2 Fig. The mindful coping model.**
(TIF)

## Author Contributions

**Conceptualization:** Meng-Li Li, Patrick Pui-Kin Kor, Justina Yat-Wa Liu.

**Investigation:** Meng-Li Li, Patrick Pui-Kin Kor, Zhi-Ying Zhang.

**Methodology:** Meng-Li Li, Patrick Pui-Kin Kor, Zhi-Ying Zhang, Justina Yat-Wa Liu.

**Supervision:** Patrick Pui-Kin Kor, Justina Yat-Wa Liu.

**Writing – original draft:** Meng-Li Li.

**Writing – review & editing:** Patrick Pui-Kin Kor, Zhi-Ying Zhang, Justina Yat-Wa Liu.

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
