## [Decision Letter · Decision Letter 0]

7 Mar 2024

PONE-D-23-43186Feasibility and preliminary effects of a Mindfulness-Based Physical Exercise (MBPE) program for community-dwelling older people with sarcopenia: A protocol for a parallel, two-armed pilot randomised controlled trialPLOS ONE

Dear Dr. Kor, 

Thank you for submitting your manuscript to PLOS ONE. After careful consideration, we feel that it has merit but does not fully meet PLOS ONE’s publication criteria as it currently stands. Therefore, we invite you to submit a revised version of the manuscript that addresses the points raised during the review process.

We look forward to receiving your revised manuscript.

Kind regards,

Meryem Merve Ören Çelik

Academic Editor

PLOS ONE

Journal Requirements:

Reviewers' comments:

Reviewer's Responses to Questions

**Comments to the Author**

1. Does the manuscript provide a valid rationale for the proposed study, with clearly identified and justified research questions?

Reviewer #1: Yes

Reviewer #2: Yes

2. Is the protocol technically sound and planned in a manner that will lead to a meaningful outcome and allow testing the stated hypotheses?

Reviewer #1: Yes

Reviewer #2: Yes

3. Is the methodology feasible and described in sufficient detail to allow the work to be replicable?

Reviewer #1: Yes

Reviewer #2: Yes

4. Have the authors described where all data underlying the findings will be made available when the study is complete?

Reviewer #1: Yes

Reviewer #2: Yes

5. Is the manuscript presented in an intelligible fashion and written in standard English?

Reviewer #1: Yes

Reviewer #2: Yes

6. Review Comments to the Author

You may also provide optional suggestions and comments to authors that they might find helpful in planning their study.

Reviewer #1: Major Revision Required

This seems a very good protocol for a worthwhile, well designed feasibility trial. However, there are some important points that need to be re-examined.

Page Lines

10 181 This statement needs rephrasing from: “… 25 per group will produce at least a small standardised effect size in a main study [46].” to something like “… 25 per group is likely to provide sufficient information for planning the proposed subsequent main randomised trial with an anticipated standardised effect size of about 0.2 [46].”

18 Primary outcomes and measures: This section needs to stipulate somewhere that a primary outcome is also the estimation of the SD for the main endpoint measure. Feasibility relates to statistical aspects of the future trial as well as practicability. See comments below regarding page 24.

23 Probably best not to conduct statistical significance tests with p-values reported, but merely report (say) the mean difference between the two randomised groups with corresponding 95% confidence intervals.

24 Reference [46] states: “When the outcome is a continuous variable, the sample size calculation requires an accurate estimate of the standard deviation of the outcome measure. A pilot trial can be used to get an estimate of the standard deviation, which could then be used to anticipate what may be observed in the main trial. However, an important consideration is that pilot trials often estimate the standard deviation (SD) parameter imprecisely.”

Thus, the main statistical issue in this feasibility study is to provide information to help ascertain a meaningful standardised effect size. This requires an estimate of the SD for the intended main outcome variable for planning the size of the subsequent randomised Phase III controlled trial. Consequently, the Statistical Analysis section should state this explicitly and thus the eventual results section requires the SD to be reported.

This in turn implies that the principal endpoint for the subsequent trial needs to be identified and clearly stipulated in this Protocol.

Some minor points are listed below:

Page Lines

9 169 ‘CONSORT’ is misspelt

11 203 “Prifysgol” is the Welsh language word for “University” so need not be included here.

24 13 Reporting minimum and maximum values rather than quartiles is likely to be more informative.

24 18-19 I cannot imagine ‘generalised estimating equations; will really be necessary.

Reviewer #2: This protocol is an intervention that addresses a very important problem that concerns public health, gerontology and geriatrics due to its magnitude and significance in the healthy aging process. Sarcopenia is a predictor of dependency, loss of autonomy, increased mortality and higher health care costs. Having scientifically verifiable intervention strategies provides an incentive for their application, even more so by including two factors as important as motivation and adherence that are recognized as problems in maintaining a longer-term program. A randomized controlled pilot trial is a good initial design to continue searching for more viable actions and their effects in the prevention of Sarcopenia.

7. PLOS authors have the option to publish the peer review history of their article (what does this mean?). If published, this will include your full peer review and any attached files.

Reviewer #1: No

Reviewer #2: **Yes: **Elva Dolores Arias Merino.

---

## [Author Response · Author response to Decision Letter 0]

12 Mar 2024

Dear Editor and Reviewers,

Thank you for your time to review our revised manuscript and your valuable comments and suggestions on it. We have made all of the revisions on the paper with careful considerations to your comments and they are described as below. The changes in the manuscript are also highlighted in yellow:

R1(1): 10 181 This statement needs rephrasing from: “…25 per group will produce at least a small standardised effect size in a main study [46].” to something like “…25 per group is likely to provide sufficient information for planning the proposed subsequent main randomised trial with an anticipated standardised effect size of about 0.2 [46].”

Response: We have adopted the suggestions and rephrased the sentence on Page 10, Line 182-184. 

R1(2): 18 Primary outcomes and measures: This section needs to stipulate somewhere that a primary outcome is also the estimation of the SD for the main endpoint measure. Feasibility relates to statistical aspects of the future trial as well as practicability. See comments below regarding page 24.

Response: We have supplemented the primary outcome on Page 21, Line 287-288. In this pilot study, the primary outcome is handgrip strength. The standardised mean difference (Conde’s d) with corresponding 95% confidence intervals will be reported to describe the mean differences between the intervention group and the control group (Page 24, Line 367-370).

R1(3): 23 Probably best not to conduct statistical significance tests with p-values reported, but merely report (say) the mean difference between the two randomised groups with corresponding 95% confidence intervals.

Response: We adopted the suggestions and revised the method section on Page 24, Line 367-370: 

The standardised mean difference with corresponding 95% confidence intervals will be presented to describe the mean differences of both primary and secondary outcomes between the intervention group and the control group [71]. All data analyses will be conducted with two-tailed tests with a significance level of p < 0.05.

R1(4): 24 Reference [46] states: “When the outcome is a continuous variable, the sample size calculation requires an accurate estimate of the standard deviation of the outcome measure. A pilot trial can be used to get an estimate of the standard deviation, which could then be used to anticipate what may be observed in the main trial. However, an important consideration is that pilot trials often estimate the standard deviation (SD) parameter imprecisely.”

Thus, the main statistical issue in this feasibility study is to provide information to help ascertain a meaningful standardised effect size. This requires an estimate of the SD for the intended main outcome variable for planning the size of the subsequent randomised Phase III controlled trial. Consequently, the Statistical Analysis section should state this explicitly and thus the eventual results section requires the SD to be reported. This in turn implies that the principal endpoint for the subsequent trial needs to be identified and clearly stipulated in this Protocol.

Response: We adopted the suggestions and revised the method section on, Page24, Line 358-361, reporting SD and Page 24, Line 367-370, stating the mean difference (see the above response #R1(3)). 

Some minor points are listed below:

Page Lines

R1(5): 9 169 CONSORT is misspelt.

Response: We have revised. Thank you! 

R1(6): 11 203 “Prifysgol” is the Welsh language word for “University” so need not be included here.

Response: We have deleted it. Thank you! 

R1(7): 24 13 Reporting minimum and maximum values rather than quartiles is likely to be more informative.

Response: We have adopted the suggestions and added the variables of minimum and maximum values on Page 24, Line 360-361. 

R1(8): 24 18-19 I cannot imagine “generalised estimating equations” will really be necessary.

Response: After consulting two statisticians, we adopted the GEE analysis because this method accounts for intra-correlated pre-test and post-test measures and accommodates missing data. Besides GEE provide robust results for non-normally distributed continuous data. 

Reviewer #2: This protocol is an intervention that addresses a very important problem that concerns public health, gerontology and geriatrics due to its magnitude and significance in the healthy aging process. Sarcopenia is a predictor of dependency, loss of autonomy, increased mortality and higher health care costs. Having scientifically verifiable intervention strategies provides an incentive for their application, even more so by including two factors as important as motivation and adherence that are recognized as problems in maintaining a longer-term program. A randomized controlled pilot trial is a good initial design to continue searching for more viable actions and their effects in the prevention of Sarcopenia.

Response: Thank you for your comments!

---

## [Decision Letter · Decision Letter 1]

1 Apr 2024

Feasibility and preliminary effects of a Mindfulness-Based Physical Exercise (MBPE) program for community-dwelling older people with sarcopenia: A protocol for a parallel, two-armed pilot randomised controlled trial

PONE-D-23-43186R1

Dear Dr. Kor,

We’re pleased to inform you that your manuscript has been judged scientifically suitable for publication and will be formally accepted for publication once it meets all outstanding technical requirements.

Kind regards,

Meryem Merve Ören Çelik

Academic Editor

PLOS ONE

Reviewers' comments:

Reviewer's Responses to Questions

**Comments to the Author**

1. Does the manuscript provide a valid rationale for the proposed study, with clearly identified and justified research questions?

Reviewer #1: Yes

2. Is the protocol technically sound and planned in a manner that will lead to a meaningful outcome and allow testing the stated hypotheses?

Reviewer #1: Yes

3. Is the methodology feasible and described in sufficient detail to allow the work to be replicable?

Reviewer #1: Yes

4. Have the authors described where all data underlying the findings will be made available when the study is complete?

Reviewer #1: Yes

5. Is the manuscript presented in an intelligible fashion and written in standard English?

Reviewer #1: Yes

6. Review Comments to the Author

You may also provide optional suggestions and comments to authors that they might find helpful in planning their study.

Reviewer #1: Accept

This seems a very good protocol for a worthwhile and well-designed study. The authors have now included the estimation of the standardised effect size for planning an eventually larger Phase III trial as an objective of this study. However, I remain concerned that the statistical analysis plan over complicates the eventual analysis that will be needed. More focus on confidence intervals rather than p-values from statistical tests of significance is likely to be required.

7. PLOS authors have the option to publish the peer review history of their article (what does this mean?). If published, this will include your full peer review and any attached files.

Reviewer #1: No

---

## [Editor Report · Acceptance letter]

8 Apr 2024

PONE-D-23-43186R1 

PLOS ONE

Dear Dr. Kor, 

I'm pleased to inform you that your manuscript has been deemed suitable for publication in PLOS ONE. Congratulations! Your manuscript is now being handed over to our production team.

Kind regards, 

on behalf of

Dr. Meryem Merve Ören Çelik 

Academic Editor

PLOS ONE